# Patients with Cardiovascular Implantable Electronic Devices in the Era of COVID-19 and Their Response to Telemedical Solutions

**DOI:** 10.3390/medicina58020160

**Published:** 2022-01-21

**Authors:** Diana Paskudzka, Łukasz Januszkiewicz, Roman Załuska, Agnieszka Kołodzińska, Łukasz Łyżwiński, Marcin Grabowski

**Affiliations:** 1First Department of Cardiology, Medical University of Warsaw, 1a Banacha Street, 02-097 Warsaw, Poland; lukasz.jan.januszkiewicz@gmail.com (Ł.J.); aa.kolodzinska@wp.pl (A.K.); luke.lyzwinski@gmail.com (Ł.Ł.); marcin.grabowski@wum.edu.pl (M.G.); 2Department of Management and Logistics in Healthcare, Medical University of Lodz, 90-131 Lodz, Poland; r.zaluska@tlen.pl

**Keywords:** COVID-19, telemedicine, CIED, telehealth, home monitoring

## Abstract

*Background and objectives:* The COVID-19 pandemic has transformed the healthcare system, leading to the rapid implementation of telemedical solutions, especially in cardiology. The aim of this survey was to evaluate the patients (pts) with cardiac implantable electronic devices (CIED) perspectives on the telemedicine elements such as teleconsultation, telemonitoring, and e-prescription. *Materials and methods:* An anonymous questionnaire was created and delivered to CIED pts who came to the ambulatory outpatient clinic. In this survey, we evaluated teleconsultation, home monitoring systems, and e-prescription in the 17 single-choice and multiple-choice questions and a rating on a scale of 0 to 10. *Results:* During the four-month period, 226 pts (58% male) completed the questionnaire. Regular visits were most frequent in pts living in the urban area where the clinic was located, and least frequent in those living in rural areas (*p* = 0.0158). Moreover, 89 pts (39%) had teleconsultation before CIED interrogation, and satisfaction was 99%; 24 pts (11%) had home-monitoring control and 135 pts (60%) would have liked to have this opportunity; 88 pts (34.5%) would be able to pay additional costs for home-monitoring, with a mean amount of 65 PLN (±68.24). The e-prescription system was used by 203 pts (90%), and it was evaluated with 8.6 points (±2) on a scale from 0 to 10 points. *Conclusions:* The COVID-19 pandemic disrupted the previous functioning of the health system, and telemedicine became an alternative to traditional ambulatory visits and proved to be essential in the continuity of patient care. There is a substantial need for further development of telemedicine solutions in the healthcare system.

## 1. Introduction

The first cases of SARS-CoV-2 infection in the world were reported in December 2019. The high infectivity rate led to a rapid spread around the world and on 11 March 2020 the World Health Organization declared a pandemic [1]. The SARS-CoV-2 virus causes a respiratory disease known as COVID-19, which demonstrates a higher mortality rate among patients with comorbidities, especially with cardiovascular disorders [2]. Due to the epidemiological situation, healthcare providers had to modify their methods of treatment in order to limit personal contact with their patients. They increasingly turned to the use of telemedicine as a solution.

In 2018, Piotrowicz et al. underlined that telemedicine is a dynamically developing field in the medical industry [3]. However, until the beginning of the pandemic, telemedical solutions were not widely implemented and regulated.

More than a year has passed since the pandemic was declared, and society has accepted many restrictions, such as social distancing and mask wearing, as part of everyday life. This has been met with mixed public reactions.

The aim of the study was to examine the patient perception of epidemiological restrictions implemented due to the COVID-19 pandemic in our cardiac implantable electronic devices (CIED) clinic. We also assessed new solutions such as teleconsultation, e-prescriptions, and home-monitoring.

## 2. Materials and Methods

An anonymous questionnaire was distributed among patients with CIEDs who were consulted in our ambulatory CIED clinic. Patients completed the questionnaire in either electronic or paper form. Data collected on paper were later entered into the database.

The questionnaire consisted of 17 questions. These were open-ended as well as multiple-choice rating questions, where possible answers ranged on a 0-to-10-point scale (where 0 meant not satisfied at all, while 10 points represented the highest possible satisfaction). The first section was related to general data about the patient, i.e., gender, approximate age, place of residence, and type of implanted cardiac device. The next section pertained to the ambulatory visit. Evaluation of teleconsultations with the clinic was the topic of the next section. Finally, patients answered questions on the home-monitoring system (automatic transmission of implanted patient device data to the physician) and their satisfaction with the use of e-prescriptions.

Responses were analyzed generally and in subgroups, i.e., by gender, type of implanted device, place of residence, and age group. In the case of statistical significance within a subgroup, the results were presented.

The continuous variables were described as mean and standard deviation (SD). The categorical variables were presented as numbers and percentages. Chi-square or tau-kendalla tests were used, as appropriate. A *p*-value of less than 0.05 was considered statistically significant.

## 3. Results

Between May 2020 and August 2020, 226 (58% male) patients completed the questionnaire. The general group characteristics are presented in Table 1.

Regular visits were most common for patients living in the city of the ambulatory clinic, which has a population of more than 1 million (65 pts, 73.9%). They were least common for those living in rural areas (21 pts; 48.8%, *p* = 0.0158). The regularity of visits and differences in rescheduling are presented in Figure 1.

Only 89 patients of those surveyed had a teleconsultation before the follow-up of the questionnaire. Almost all of these patients were satisfied with the teleconsultation performed (88 of 89 pts, 98.9%). Details of the teleconsultation are presented in Figure 2.

Patients with cardiac resynchronization therapy devices, compared to implantable cardioverter-defibrillator or pacemaker groups, expressed the lowest desire for the universal application of teleconsultation (45.5% versus 83.7% and 76.2%; *p* = 0.013).

Only a few of the respondents were already in the remote monitoring group, but more patients answered that they would like to be monitored remotely. However, if additional fees would apply, this number dropped to 78 patients (34.5%) who were able to cover the additional costs. The average amount per month declared as acceptable for this service was 65 zloty (±68.24 zloty/14.38 ±15.1 euro). Patients living in rural areas (farthest from the clinic) declared a higher acceptable amount (85.83 ± 74.28 zloty/18.99 ± 16.44 euro). The age group above 80 years old also declared a higher amount (71.54 ± 62.96 zloty/15.83 ± 13.93 euro). Detailed data are presented in Figure 3.

The great majority of the patients used e-prescriptions during the COVID-19 pandemic. This experience was rated at an average 8.6 points (±2) on a scale of 0 to 10 points, where 0 meant not satisfied at all and 10 meant fully satisfied. Details are presented in Figure 4.

Functionality of e-prescriptions was rated highest by the 51–60 age group (9.2 ± 1.44; *p* = 0.0304) and was rated higher by females (8.86 ± 1.86; *p* = 0.0314). The questionnaire and detailed results of the statistical analyses are included in the Appendix A.

## 4. Discussion

The COVID-19 pandemic has created new challenges, disrupting existing medical practice. Telemedicine has become an alternative to a significant portion of previous outpatient visits and has proven to be a necessary clinical innovation [4,5].

One method to maintain epidemiological restrictions and to limit face to face visits while providing care for patients with CIEDs is remote monitoring. Only 11% of patients among the respondents were in the remote monitoring program, more than half of the patients indicated a preference for this method. Currently, all patients under remote monitoring care are not required to pay any fees for this service.

Remote patient monitoring has been validated as a safe alternative to in-clinic visits worldwide. This type of continuous monitoring facilitates a rapid response to sudden clinical and technical device problems, often much sooner than for patients monitored traditionally. This leads to improved patient outcomes and lower costs of care [6,7,8,9]. Miller et al. pointed out that the COVID-19 pandemic initiated a digital breakthrough in medicine. A return to the previous model of treatment based on in-person visits will likely no longer be viable, and the future will see a hybrid of inpatient care supplemented by continuous monitoring in the patient’s home via implantable and/or wearable devices [10]. Auricchio et al. argue to consider systematic activation of the remote monitoring function at implantation or, through default programming, in all cardiac rhythm monitoring devices. This would allow for easy activation of the remote monitoring function without the need for a physical visits by the patient, especially when access to the outpatient clinic is critically limited, such as during a pandemic and/or when human resources are limited [11]. E-prescription as part of the telehealth system were received with a very positive response by patients. They improved the quality of care and patient safety, i.e., by reducing prescription errors [12]. They also reduced the number of patient visits for refill prescription, freeing up schedules and shortening waiting times for other outpatient visits. In addition, when a prescription is sent electronically to a pharmacy, it also reduces the waiting time for medications and the duration of an in-person visit to the pharmacy because the order can be prepared in advance. In addition, the implementation of e-prescriptions minimizes the risk of being exposed to a virus compared with using paper [13].

It is difficult to compare perceptions of telemedicine before and during a pandemic. For patients who used telemedicine services before the COVID-19 epidemic, it was usually their own choice. For telemedicine during the pandemic, in most cases it was a necessity—there was no other way to contact medical staff. Holtz compared patients’ experiences of using telemedicine. All participants had a generally good experience. The pre-pandemic telemedicine users preferred this mode of communication when they were too ill to leave home, when it was not an urgent condition, and when clinics were unavailable. This group was more likely to disagree that they would receive better care in person and would like to personally visit. Patients using telemedicine for the first time during the pandemic were more likely to report using these services to avoid waiting rooms and reduce their risk of infection. They had a higher perceived care by a physician and less worries about continuity of treatment [14].

### Limitations

This is a single-center study from a level three reference hospital with a relatively small sample size, making data generalization difficult. Despite these limitations, this is one of the first studies on the CIED patients’ assessment of telemedical solutions

## 5. Conclusions

Telemedicine has made great advances and found widespread use from the onset of the COVID-19 pandemic, and it is likely to continue to play an important role in healthcare. The main message delivered by patients regarding telemedicine solutions is positive, with high rate of satisfaction with teleconsultation and a relevant need for this healthcare form. However, further legal and ethical regulations are needed. We may be at a moment where telemedicine will become one of the pillars of the modern healthcare system.

## Figures and Tables

**Figure 1 medicina-58-00160-f001:**
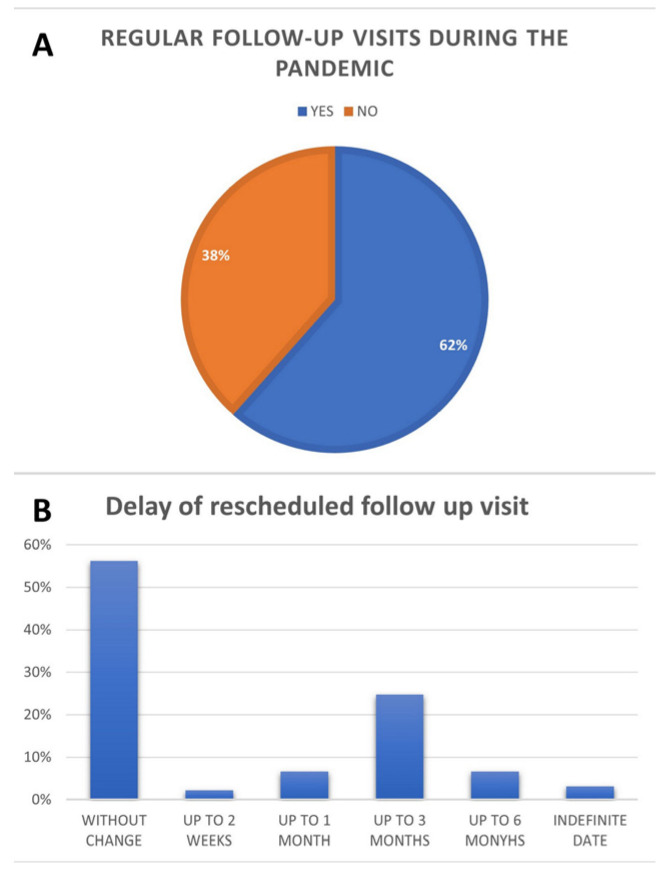
Change date of follow-up visits during the pandemic. (**A**) Regular follow-up visits during the pandemic; (**B**) Delay of rescheduled follow up visits.

**Figure 2 medicina-58-00160-f002:**
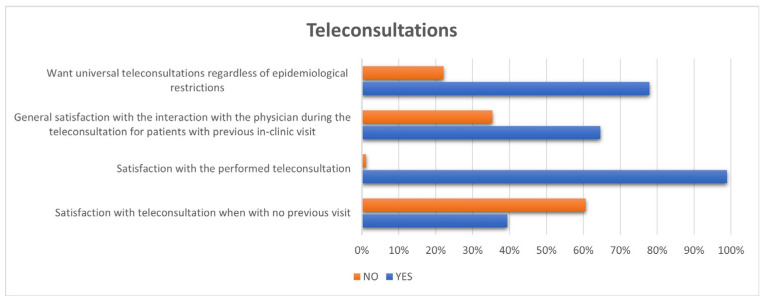
Assessment of teleconsultation.

**Figure 3 medicina-58-00160-f003:**
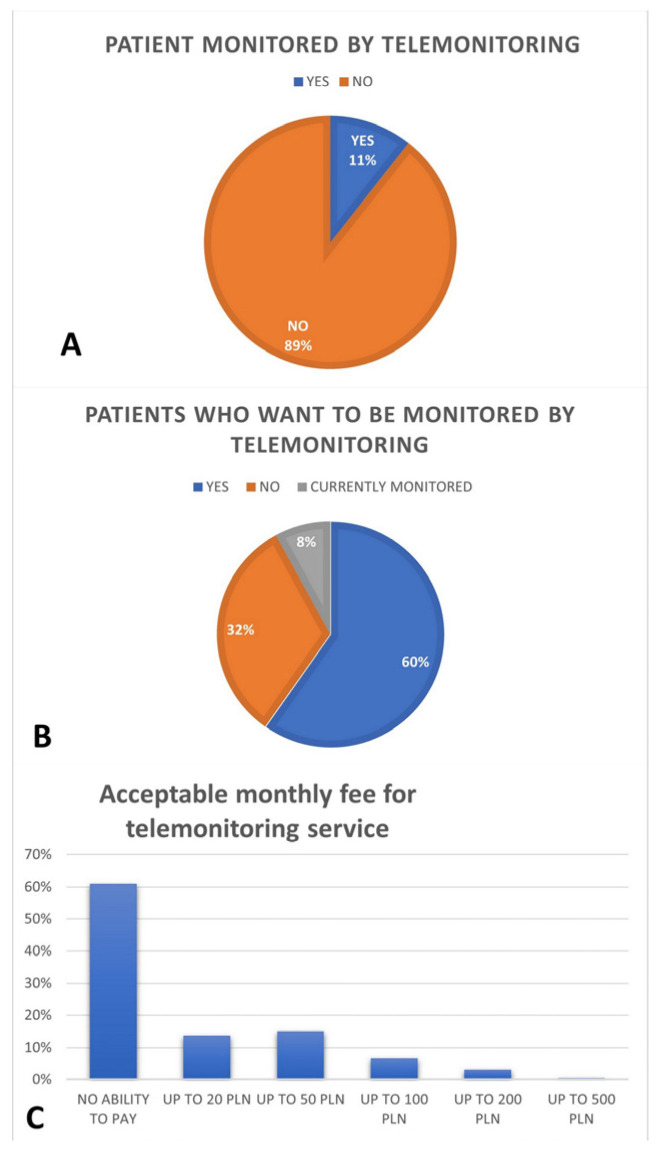
Characteristics of the home-monitoring system. (**A**) Patients monitored by telemonitoring; (**B**) Patients who want to be monitored by telemonitoring; (**C**) Acceptable monthly fee for telemonitoring service.

**Figure 4 medicina-58-00160-f004:**
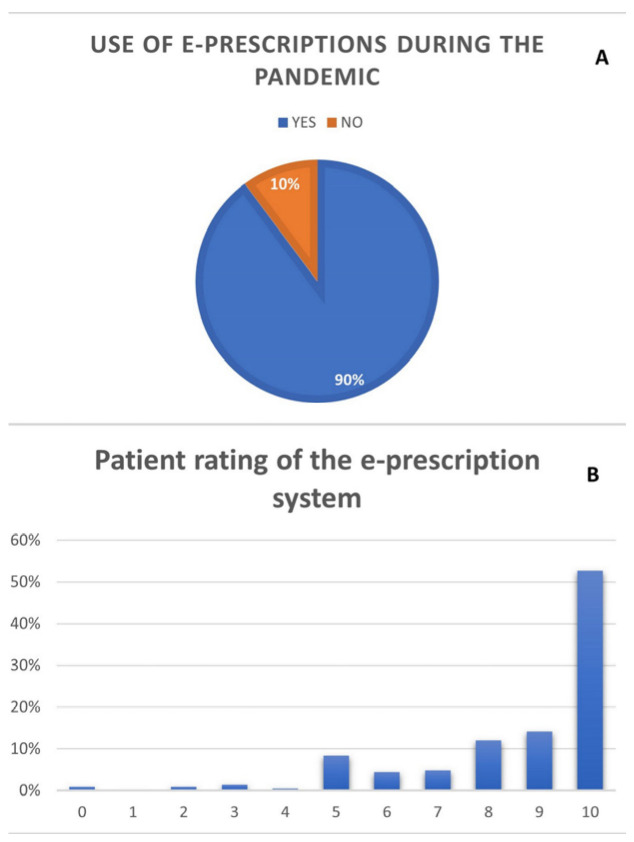
Details of e-prescription system. (**A**) Use of e-prescriptions during the pandemic; (**B**) Patient rating of the e-prescription system, score between 0 to 10 points, where 0 is the lowest and 10 is the highest grade.

**Table 1 medicina-58-00160-t001:** Patient’s characteristics. Data are presented as number (percentage) of patients.

Patient, *n* (%)	226 (%)
Male gender	131 (58)
Age group	
>80 y	50 (22.1)
71–80 y	43 (19)
61–70 y	47 (20.8)
51–60 y	25 (11)
41–50 y	29 (12.9)
31–40 y	22 (9.8)
<30 y	10 (4.4)
Accommodation	
City > 1 million inhabitants	88 (39)
City > 100,000 and <1 million inhabitants	35 (15.5)
City < 100,000 inhabitants	60 (26.5)
Rural area	43 (19)
Implantable device type	
Pacemaker	122 (54)
Implantable cardioverter-defibrillator	92 (40.7)
Cardiac resynchronization therapy	11 (4.9)
Implantable loop recorder	1 (0.4)

*n*, number; y, years old.

## Data Availability

The data presented in this study are available upon reasonable request from the corresponding author.

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
