# Peer review of "Patients with Cardiovascular Implantable Electronic Devices in the Era of COVID-19 and Their Response to Telemedical Solutions"

_medicina, 2022, doi:10.3390/medicina58020160_

Round 1

Reviewer 1 Report

Pandemics pose unique challenges to health care delivery. Patients presenting for in-person care who screen positive for high-risk features should be isolated immediately to avert further contact with patients and health care workers. Telemedicine visits  at home, greatly limiting travel and exposure and permitting uninterrupted care of established patient so it t is  appropriate in minimizing the risk of COVID-19 transmission. This retrospective assessment of a relatively large group of patients is an important contribution to the ongoing discussion on the use of telemedicine in a pandemic. It would be interesting to learn how the objective value of teleconsultations looked like (e.g. whether it was possible to detect emergency situations with the help of televisits and necessary immediate interventions, etc.). Defining selected clinics as highest in Poland could be met with good criticism of others, having an equally high level of reference centers, this phrase should be removed.

Author Response

Response to Reviewer 1 Comments

Point 1: Pandemics pose unique challenges to health care delivery. Patients presenting for in-person care who screen positive for high-risk features should be isolated immediately to avert further contact with patients and health care workers. Telemedicine visits  at home, greatly limiting travel and exposure and permitting uninterrupted care of established patient so it is appropriate in minimizing the risk of COVID-19 transmission. This retrospective assessment of a relatively large group of patients is an important contribution to the ongoing discussion on the use of telemedicine in a pandemic. It would be interesting to learn how the objective value of teleconsultations looked like (e.g. whether it was possible to detect emergency situations with the help of televisits and necessary immediate interventions, etc.).

Response 1: Thank you for this significant comment. Unfortunately, since the questionnaire was anonymous, we did not have access to any sensitive data regarding the direct results of teleconsultation. We may, however, assess it in the future studies measuring the actions taken during teleconsultation.

Point 2: Defining selected clinics as highest in Poland could be met with good criticism of others, having an equally high level of reference centers, this phrase should be removed.

Response 2: Thank you for your note. This phrase was removed from the manuscript.

Reviewer 2 Report

Please check the text in Table Nr.1.

It is not clear what the authors want to say in the conclusions. The conclusions need to be revised - they need to be more accurate and concrete, relevant to results.

Author Response

Response to Reviewer 2 Comments

Point 1: Please check the text in Table Nr.1.

Response 1: Thank you for your note. I corrected the misspellings.

Point 2: It is not clear what the authors want to say in the conclusions. The conclusions need to be revised - they need to be more accurate and concrete, relevant to results.

Response 2: Thank you for this significant comment. I added the comment that the main message delivered by patients regarding telemedicine solutions is positive, with high rate of satisfaction with teleconsultation and a relevant need of this healthcare form.

Reviewer 3 Report

Dear authors, the work is very nice, but you should address a deep revision of the objective of the paper.

I think the conclusion is too basic for a new paper: telemedicine is a good alternative to outpatient visits and contributes to improving patient care.

You should try to provide more relevant information and you should also remark the originality of your work.

Author Response

Response to Reviewer 3 Comments

Point 1: Dear authors, the work is very nice, but you should address a deep revision of the objective of the paper. I think the conclusion is too basic for a new paper: telemedicine is a good alternative to outpatient visits and contributes to improving patient care. You should try to provide more relevant information and you should also remark the originality of your work.

Response 1:.   Thank you for this comment. I changed the conclusion. I added the comment that the main message delivered by patients regarding telemedicine solutions is positive, with high rate of satisfaction with teleconsultation and a relevant need of this healthcare form.

Round 2

Reviewer 2 Report

After corrections, the publication corresponds to the level of the scientific article. Thank you!

Author Response

Dear Reviewer, thank you for your comment.

Reviewer 3 Report

Dear author, your research work is very nice, and it is very well written but the conclusion is nothing new. It is already known that telemedicine in times of pandemic gives very good results.
In order to improve your work, it should be completed, for example by providing data prior to the pandemic, in order to study the possible differences between both periods of time: pre and post pandemic

Author Response

Response to Reviewer 3 Comments

Point 1: Dear author, your research work is very nice, and it is very well written but the conclusion is nothing new. It is already known that telemedicine in times of pandemic gives very good results.

In order to improve your work, it should be completed, for example by providing data prior to the pandemic, in order to study the possible differences between both periods of time: pre and post pandemic

Response 1: Dear Reviewer, thank you for your comments. We agree that telemedicine in times of pandemic gives very good results. Unfortunately, since the survey was carried out during pandemic time only, we are not able to provide their results from pre-pandemic time. Therefore, in the discussion we cited an article comparison of telemedicine before and during a pandemic. It was added that “It is difficult to compare perceptions of telemedicine before and during a pandemic. For patients who used telemedicine services before the COVID-19 epidemic, it was usually their own choice. For telemedicine during the pandemic, in most cases it was a necessity - there was no other way to contact medical staff. Holtz compared patients' experiences of using telemedicine. All participants had generally good experience. The pre-pandemic telemedicine users preferred this mode of communication when they were too ill to leave home, but weren’t an urgent condition and when clinics were unavailable. This group was more likely to disagree that they would receive better care in person and would like to personally visit. Patients using telemedicine for the first time during the pandemic were more likely to report using these services to avoid waiting rooms and reduce their risk of infection. They had a higher perceived care by a physician and less worries about continuity of treatment [14]”. New reference was added.
